# The Role of Vitamin C in Two Distinct Physiological States: Physical Activity and Sleep

**DOI:** 10.3390/nu12123908

**Published:** 2020-12-21

**Authors:** Aneta Otocka-Kmiecik, Aleksandra Król

**Affiliations:** Department of Experimental Physiology, Medical University of Lodz, 92-215 Lodz, Poland; aneta.otocka-kmiecik@umed.lodz.pl

**Keywords:** vitamin C, oxidative stress, physical activity, performance, cellular adaptation, sleep, insomnia, obstructive sleep apnea, restless legs syndrome

## Abstract

This paper is a literature overview of the complex relationship between vitamin C and two opposing physiological states, physical activity and sleep. The evidence suggests a clinically important bidirectional association between these two phenomena mediated by different physiological mechanisms. With this in mind, and knowing that both states share a connection with oxidative stress, we discuss the existing body of evidence to answer the question of whether vitamin C supplementation can be beneficial in the context of sleep health and key aspects of physical activity, such as performance, metabolic changes, and antioxidant function. We analyze the effect of ascorbic acid on the main sleep components, sleep duration and quality, focusing on the most common disorders: insomnia, obstructive sleep apnea, and restless legs syndrome. Deeper understanding of those interactions has implications for both public health and clinical practice.

## 1. Introduction

Restful sleep and intense physical activity are two behaviors mediated by completely different physiological mechanisms.

During acute exercise sympathetic activity of the autonomic nervous system is stimulated. A rise in catecholamines and an increase in production and catabolism of cortisol occurs. Cardiac output, stroke volume, heart rate, and blood pressure increase. Also, pulmonary ventilation, tidal volume, and breathing rate rise. This leads to higher oxygen uptake and carbon dioxide removal. Maximal oxygen uptake may be limited mainly by the cardiovascular system but also by a low level of mitochondrial enzymes in skeletal muscles. Lactate production is increased, especially during vigorous exercise, with increased reliance on glycolysis. The temperature in the muscles rises, muscle fatigue increases [1].

Sleep, in turn, is a physiological rest state for the body and the mind, regulated in part by control nuclei of the autonomic nervous system. Three types of sleep are distinguishable in all mammals: wakefulness, Non-Rapid Eye Movement (NREM) sleep, and Rapid Eye Movement (REM) sleep, and the regulation of the most important physiological functions differs according to the states of sleep and wakefulness [2].

In transition from the waking state to drowsiness and into sleep, sympathetic tone decreases, the respiratory rate slows down, and regular breathing, helping to promote gas exchange, is induced. At the same time, vagal tone increases. Characteristic for NREM sleep, a state of parasympathetic dominance is a reduction in heart rate and cardiac output. Peripheral vascular resistance and blood pressure are also reduced. Metabolic heat production and body temperature are lowered, to minimize energy expenditure. During REM stage, skeletal muscles, except for the eyes and the diaphragmatic breathing muscle, are atonic and immobile, to inhibit body movement. Parasympathetic tone dominates during tonic REM, while throughout phasic REM sympathetic tone increases, as sympathovagal balance reverses. As a consequence, blood pressure and heart rate rise dramatically and are more varied, similar to the daytime pattern. The respiratory rhythm is irregular [2,3].

Nevertheless, the clinically important relationship between different aspects of sleep and physical activity has been confirmed as a subject of much research. Moreover, the association of sleep and physical activity is likely bidirectional. Sleep is not only an integral part of recovery and an adaptive process between bouts of exercise, but its increased duration and quality improves physical performance. Adequate sleep is vital for tissue repair and muscle growth, particularly following exercise [4]. On the other hand, moderate exercise exerts a beneficial effect on the quality of sleep by decreasing sleep latency, reducing daytime sleepiness, increasing total sleep time and slow-wave sleep, which aids in the recovery process [5,6,7]. Furthermore, regular physical activity is associated with reduced risk of having disturbed sleep and decreased obstructive sleep apnea severity [8].

Both physical activity and sleep change the prooxidant–antioxidant balance in the body. Acute exercise is responsible for escalating production of highly reactive oxygen species (ROS) by increasing mitochondrial oxygen consumption. The imbalance between ROS production and antioxidant systems may lead to oxidative damage to proteins, lipids, and DNA. The antioxidant protection in the form of endogenous and exogenous molecules is launched. However, it may not be enough to fight the deleterious effects of oxidative stress. Either excessive production of ROS at acute exercise or depletion of antioxidant systems calls for the need to boost up the endogenous pathway and supplement the exogenous antioxidants.

Sleep gives the opportunity to restore the endogenous antioxidant mechanisms. One of its roles is to increase the organism’s resistance to oxidative stress, while sleep loss/deprivation induces oxidative stress and reduces anti-oxidant response. Sleep promotes removal of free radicals accumulated during wakefulness and reduces ROS levels in the brain [9]. There is a bidirectional relationship between sleep and oxidative stress. Sleep acts as an antioxidant for the body and the brain; on the other hand, oxidative stress helps to induce sleep [10].

In exercise and sleep, regardless of their contradictory effects on the redox state, supplementation with exogenous antioxidants such as vitamins C and E may theoretically be advantageous. In exercise they may aid to fight the deleterious effects of oxidative stress, in sleep they may help restore the prooxidant–antioxidant balance.

The aim of this review is to

(a)discuss the role of vitamin C and vitamins C and E in exercise metabolism, which has important implications in the primary and secondary prevention of insulin resistance, diabetes, metabolic syndrome, and obesity;(b)summarize the results of studies on antioxidant and anti-inflammatory properties of vitamin C and vitamins C and E in exercise;(c)address the influence of vitamin C and vitamins C and E on the regular exercise-induced increase in performance in athletes and recreationally active men;(d)compare the effects of vitamin C and vitamins C and E supplementation on exercise in two distinct groups: the young and the elderly;(e)discuss the association between vitamin C intake and sleep symptoms;(f)summarize the existing knowledge on the relationship of vitamin C and different physiological and psychological sleep disorders: insomnia, obstructive sleep apnea, and restless legs syndrome;(g)determine the benefits of vitamin C supplementation for sleep health.

## 2. Physical Activity

As early as 600 B.C., Susruta (Sushruta), a physician from India, was wise enough to prescribe daily exercise of moderate intensity for prevention and treatment of diseases, notably to fight the effects of diabetes and obesity. At about 400 B.C. Hippocrates of Kos provided a written description of the benefits of exercise. He prescribed “early-morning walks to reduce [the body], and render the parts about the head light, bright and of good hearing, while they relax the bowels”. Nowadays, regular exercise is a known factor that prevents the development of certain civilization diseases, such as obesity, type 2 diabetes, metabolic syndrome, non-alcoholic fatty liver disease, atherosclerosis, coronary artery disease, certain malignancies, and others. Yet, *Physical Activity Guidelines for Americans*, 2nd edition, shows that nearly 80% of adults are not meeting the key guidelines for both aerobic and muscle-strengthening activity, while only about half meet the key guidelines for aerobic physical activity [11].

However, exercise is not always beneficial. A single prolonged or high-intensity physical activity leads to increased oxygen flux and excessive release of reactive oxygen species (ROS), such as superoxide anion (O˙_2_), peroxyl (RO˙_2_), and hydroxyl radical (˙OH).) [12,13]. This is accompanied by excessive formation of reactive nitrogen species (RNS) derived from nitric oxide such as peroxynitrite [13]. This increase in free radicals formation during exercise was first evidenced in 1978 [14], where an hour of exercise on a bicycle ergometer at 50% of maximal oxygen consumption (VO_2_max) led to an increase in pentane, an indicator of lipid peroxidation measured in expired air. Further studies confirmed that physical activity leads to lipid peroxidation [15,16].

The oxidative stress, which occurs when the antioxidant capacity is exceeded, leads to various changes in the molecular structure of lipids, proteins, glutathione, and DNA. These changes result in cell injury and activate cell death signalling cascades. In untrained individuals, prolonged exercise of moderate intensity or even short-term exercise of high intensity may lead to muscle damage. As inflammation develops, neutrophils are activated. These changes manifest as the delayed onset of muscle soreness and reduction in performance in the days after intensive training [17]. In blood, an increase in biomarkers of muscle damage, such as creatine kinase (CK) [17], lactate dehydrogenase (LDH), as well as acute phase response measured as C-reactive protein (CRP), is observed [18].

## 3. Antioxidant Defense

A whole array of endogenous antioxidants is launched in order to fight the deleterious effects of oxidative stress. Enzymatic antioxidants are released, i.e., superoxide dismutase (SOD), glutathione peroxidase (GPx), catalase (CAT), and paraoxonase 1 (PON1). Besides these enzymes, a number of other, nonenzymatic particles are also found in the organism, i.e., glutathione, bilirubin, uric acid, coenzyme Q, α-lipoic acid, and ferritin. Some exogenous substances, such as phytochemicals (polyphenols, carotenoids, etc.) and vitamins (α-tocopherol, β-carotene, and ascorbic acid) also serve an antioxidant purpose [19]. The exact role of these antioxidants in exercise is not fully recognised. There are numerous studies searching for mechanisms that would enhance adaptation to exercise when training. In particular, ways to stimulate the production and activity of antioxidant enzymes or generation and release of other endogenous antioxidants have been examined. Other studies are focusing on supplementation of extrinsic factors, which could enhance the body’s ability to fight the deleterious effects of oxidative stress during exercise. In this respect, vitamin C, vitamin E, and selenium are thought to be promising agents.

Therefore, supplements containing various antioxidants, especially vitamins C and E, are nowadays widely used in hope of achieving higher endurance training effects [20]. There is some evidence that supplementation with these vitamins may have a beneficial effect during training. Shafat et al. showed that implementing with these antioxidants offered protection against contraction force loss [21]. Muscle soreness was attenuated by vitamin C intake in a double-blind, randomised, crossover study [22]. Inflammatory markers (CRP, IL-6, but not TNF-α) were lower in subjects using antioxidants (multivitamins, vitamins C and E, β-carotene), regardless of the physiological activity level [23]. Adding vitamin C to a four-week training protocol of female athletes hampered the increase of creatine kinase [24]. Antioxidant supplementation can also cause an increase in erythrocyte glutathione peroxidase and glutathione values and a decrease in lipid peroxidation during intensive training [25]. Apart from the reports on the positive effect of vitamins C and E in exercise, accumulating evidence speaks out against, rather than in favour of, the efficacy of these antioxidant supplements [26].

In this part of the review we summarize the existing evidence on the effect of vitamin C administration to subjects performing physical activity, as presented in Table 1. Very often in the studies on physical activity two antioxidants, vitamins C and E, are supplemented together in the hope of achieving a synergistic effect. There are very strong premises for this approach, as vitamin C serves as an electron donor to vitamin E radicals generated in the cell membrane during oxidative stress [27]. It may, however, be misleading, as it is difficult to say if the results should be assigned to vitamin C or E, or to their common interaction. Therefore, we summarize the integrated effect of vitamins C and E in a separate table (Table 2).

## 4. Effect of Vitamins C and E on Cellular Adaptations to Exercise

The effects of antioxidant vitamin C supplementation on mitochondrial biogenesis and the efficacy of training are controversial.

Adding vitamin C alone to eight weeks aerobic training of untrained men resulted in blunting of the effects of training, as the supplemented group had no improvement of maximal oxygen uptake (VO_2_max) with a tendency to lower levels [28]. In the same study, adding vitamin C to a study protocol of three weeks rat training hindered the adaptation to training of antioxidant enzymes MnSOD and GPx in the skeletal muscles. Both of these enzymes work by reducing H_2_O_2_ to a less harmful molecule. Additionally, they prevent activation of mitochondrial biogenesis in skeletal muscle by hampering the exercise-induced expression of key transcription factors, such as PGC-1α, nuclear respiratory factor 1, and mitochondrial transcription factor A. Another observation that vitamin C is detrimental to exercise-induced skeletal muscle adaptation and mitochondrial biogenesis comes from a study on rats [29]. Here, treatment with vitamin C did not affect DNA replication in skeletal muscle but blunted the protein synthesis rate.

In addition, four weeks of aerobic training resulted in an increment in expression of peroxisome proliferator-activated receptor Ɣ (PPARƔ) and its coactivators PGC-1α and PGC-1β in a group receiving a placebo [33]. Increase in the expression of PPARƔ increases insulin sensitivity. Through this mechanism physical exercise may be helpful in preventing type 2 diabetes. However, this beneficial effect of exercise was abrogated by pretreatment with vitamins C and E. Furthermore, antioxidant supplementation also hampered endogenous antioxidant defense (superoxide dismutase (SOD1 and SOD2) and glutathione peroxidase (GPx)), regardless of the training status of the subjects.

In a double-blind, randomised, controlled trial Paulsen et al. found that adding vitamins C and E to a 10-week endurance training protocol blunted the upregulation of mitochondrial marker cytochrome c oxidase subunit IV (COX 4) (important for muscular endurance). They hampered the increase in cytosolic (but not whole muscle) PGC-1α in m. vastus lateralis, which was seen in the placebo group [34]. Also, mRNA levels of signalling proteins (nitrogen-activated protein kinase 1 (MAPK1) and CDC 24) were lower in the supplemented group. In conclusion, cellular adaptations were hampered by vitamins C and E in the exercised muscles. The possible mechanism behind this is that supplying exogenous antioxidants inhibited the production of ROS and RNS and redox-sensitive signalling.

However, another study showed conflicting results, as no blunting effect of eight weeks supplementation of vitamins C and E on skeletal muscle oxidative stress, COX4 protein, and PGC-1α mRNA was found after physical training in healthy young males [35]. Furthermore, in this study, citrate synthase (CS) (an enzyme that correlates with mitochondrial content and oxidative capacity in human skeletal muscle activity [43]) also remained unaffected by the supplementation. However, vitamins C and E were found to attenuate the protein abundance of mitochondrial transcription factors A (TFAM) and skeletal muscle superoxide dismutase (SOD2), which are downstream targets of PGC-1α gene expression. Therefore, it seems that the antioxidant supplementation affected the posttranscriptional modification. Possibly, a negative feedback mechanism is involved in these changes, where increased level of exogenous antioxidants may have resulted in a lower need of adaptation of endogenous antioxidants after exercise. Superoxide scavenging properties of vitamin C may have increased competition with SOD and reduced the requirement for SOD in response to endurance training [35].

Additionally, at high levels, exogenous antioxidants can promote oxidation, contributing to acute decrements in exercise performance. For example, vitamin C can react with metal ions released from exercise-induced tissue damage, giving rise to harmful hydroxyl radicals [44].

Yet some authors have not observed this suppressing effect of vitamin C on natural adaptations to physical training. Yfanti et al. noted no difference in the improvement in protein content of MnSOD, β-hydroxyacyl-CoA dehydrogenase, CS, and glycogen concentration in response to 12 weeks of training in a group treated with the antioxidant and in the placebo group [45]. No effect of vitamin C and E supplementation on the training-induced adaptive responses of muscle mitochondria in rats was found [46]. Protein levels of SOD, PGC-1α, and mitochondrial proteins (cytochrome oxidase I and IV, CS, ATP synthase, succinate-ubiquinone oxidoreductase, NADH-ubiquinone oxidoreductase, and long-chain acyl-CoA dehydrogenase) were not affected by the antioxidants. Bloomer et al. found no effect of vitamin C and mixed tocopherols, tocotrienols in attenuating markers of skeletal muscle injury and oxidative stress in well-trained men [47].

Some observations have shown that vitamin C supplementation resulted in significant improvement in glucose metabolism, or decrease in blood pressure, triglycerides and LDL-C (low-density lipoprotein cholesterol) [48]. Supplementation with 500 mg of vitamin C led to beneficial changes, such as a decrease in triglyceride level and an increase in HDL-C, and lowered waist circumference after training [30]. Unfortunately, it did not cause any further reduction of weight in patients with metabolic syndrome at 12 weeks endurance training. Another study did not confirm this report, as Ryan et al. observed an increase in HDL-C after endurance training regardless of vitamin C and E supplementation [49]. Also, no effect of training was seen in other blood lipids.

Furthermore, Ristow et al. found that vitamins C and E prevented favourable effects of training in previously trained and untrained individuals, such as improvement in insulin sensitivity measured as an increase in expression of reactive oxygen species-sensitive transcriptional regulators of insulin sensitivity (PPARƔ, and its coactivators, PGC-1α and PGC-1β) [33].

The aforementioned studies on the role of vitamin C in cellular adaptations to physical activity have important implications in the primary and secondary prevention of insulin resistance, diabetes, metabolic syndrome, and obesity. In the context of sleep pathologies these observations are orientated towards patients with obstructive sleep apnea as these disease entities are related.

## 5. Antioxidant and Anti-Inflammatory Properties of Vitamin C and Vitamins C and E in Exercise

Here we mention studies on the antioxidant and anti-inflammatory effects of vitamin C and E supplementation on physical activity, which are especially relevant in primary and secondary prevention of inflammatory diseases leading to sleep disorders, such as insomnia in cases of cancer or restless legs syndrome in patients undergoing hemodialysis.

Dietary antioxidants, such as vitamins C and E, are required in small amounts to work together with endogenous antioxidants to maintain redox homeostasis. However, supplemented in high doses exogenous antioxidants can inhibit signalling pathways of endogenous antioxidants and induce further oxidation, contributing to decrements in exercise performance [50].

Supplementation with vitamin C and N-acetylcysteine after an acute muscle injury induced by eccentric exercise caused an increase in markers of oxidative stress and lipid peroxidation [51]. Also, administration of vitamins C and E together to half and full Ironmen triathletes resulted in increased oxidative stress [52]. Vitamins C and E prevented the training-induced upregulation of antioxidant enzymes, such as CuZnSOD, MnSOD, and GPx1, in trained and untrained individuals [33]. Theodorou et al. also failed to find proof for any effect of vitamin C on redox homeostasis in an eccentric exercise model [37]. Furthermore, Yfanti et al. found that 12 weeks bicycle training led to an increase in oxidative stress assessed as plasma protein carbonyls concentration only in the vitamins C and E but not the placebo group. The level of lipid peroxidation was higher in a group supplemented with these antioxidants. These findings speak for a pro-oxidant rather than an antioxidant effect of vitamins C and E related to exercise training [36].

Only few data suggest antioxidant supplementation can attenuate exercise-induced tissue damage and inflammatory response. However, these data are of low quality. The study of Shaw and colleagues observed that adding vitamin C-enriched gelatin to an intermittent exercise program improves collagen synthesis and can play a beneficial role in injury prevention and tissue repair. Nevertheless, this effect may be due to gelatin amino acids rather than vitamin C [53]. Vitamin C supplementation had no beneficial effect on inflammation parameters, including CK levels, myoglobin, muscle soreness, and function after a 90-min 5% downhill run at 75% VO_2_max [54], or a 90-min intermittent shuttle run [32], suggesting no effect on free radical formation and exercise-induced inflammatory response.

As it had been suggested that part of the failure of antioxidant supplementation to reduce oxidative stress and promote health is due to administering these factors to humans with normal levels of antioxidants, Paschalis and Theodorou performed a study comparing the effect of vitamin C supplementation in subjects with high or low initial vitamin C concentration. Only in individuals with a low vitamin C initial concentration does supplementation decrease oxidative stress and may increase exercise performance [55]. Vitamin C levels are limited by the uptake from the intestines through the sodium-dependent vitamin C transporter (SVCT) 1 and 2. Therefore, many studies have shown a modest effect of vitamin C supplementation on its levels. There may be substantial differences in the absorption of this antioxidant depending on its plasma level [56]. Levine and co-workers in research on the relationship between vitamin C doses and steady-state plasma concentration showed a steep sigmoidal relationship, and proved that complete plasma saturation occurred at 1000 mg daily, while a 200 mg dose produced 80% saturation of plasma. It should also be emphasized that Levine’s studies showed that cells saturate at lower doses than plasma, probably because active transport is required for vitamin C accumulation in cells. The dose of ascorbate that produces a plasma concentration of approximately 60 µmol/L, which is needed to achieve VO_2_max of active transport, is 100 mg/d. Consistently, to produce 80 µmol/L, 1000 mg/d dose is required [57,58].

## 6. TheIinfluence of Vitamin C and Vitamins C and E on Regular Exercise-Induced Increase in Performance in Athletes and Recreationally Active Men

Performance outcomes are the measures most important for athletes. Striving to improve performance is the actual reason why supplements such as vitamin C are commonly used, but so far as to their effectiveness, the outcome measures found in scientific literature are not very promising. On the contrary, many studies supply prove that a high dosage of vitamin C has a negative rather than positive effect on muscle adaptation to exercise.

Data from animal studies suggest that deficiency of vitamin C causes weight loss and fiber atrophy in skeletal muscle [59]. However, in the human model, changes in VO_2_max did not differ between the groups receiving vitamin C or the placebo [28]. Similarly, training-induced increments in VO_2_max and running performance were not affected by the supplementation with large doses of vitamins C and E, yet the supplementation blunted the increase in markers of mitochondrial biogenesis following endurance training [28]. Roberts et al. showed that daily oral consumption of 1 g of vitamin C during a four-week high-intensity interval training period of recreationally active males does not impair training-induced improvements, such as VO_2_max, running economy, and 10 km time trial [31]. Similarly, it does not influence the decrease in mean carbohydrate and increase in mean fat oxidation rates observed during submaximal exercise. Also, no effect of vitamins C and E on performance in soccer players during the precompetitive period was found [60]. Eccentric exercise training for four weeks employed to induce extensive increases in oxidative stress failed to support any effect of vitamin C (1 g/day) supplementation in terms of muscle damage and muscle performance [37]. Muscle adaptations after exercise training, such as increased baseline muscle torque and muscle resistance to damage, were not improved by supplementation with these vitamins. The study also assessed the influence of this mixed antioxidant supplementation on blood and muscle redox status as well as on hemolysis and found no difference from the placebo. Similar results were obtained in the other studies [61].

In an animal model, a high dosage of vitamin C supplementation resulted in a lower increase in endurance performance after training than in a group receiving placebo treatment [28]. Additionally, the study showed that markers of mitochondrial synthesis, PGC-1α, increased only in the placebo group. Supplementation with high dosages of vitamin C attenuated hypertrophy of overloaded plantaris muscles in rats [62]. Moreover, some study results indicated that antioxidant supplementation can have a negative impact on strength training [38]. In particular, it hampered certain strength increases (biceps curl) and protein synthesis, although it did not affect the increase in muscle mass.

Antioxidant intake (vitamins C and E) also seems to have no effect on altitude training. In one study no difference was found between the rich and normal antioxidant diet groups of elite athletes regarding changes in hemoglobin mass, VO_2_max, or swimming performance during altitude training. However, hemoglobin concentration increased more in the antioxidant-treated group (effect size = 0.7; *p* = 0.045) with a concomitantly larger decrease in plasma and blood volumes compared to the control group. Changes in ferritin and erythropoietin from pre- to post-altitude did not differ between the groups [63]. Other authors pointed out that ingesting an antioxidant cocktail prior to exercise is likely to disrupt the delicate balance between pro- and antioxidant forces, which negatively impacts ventilation, blood lactate, economy, perception of fatigue, and performance in young healthy males [42].

There is very little proof of a beneficial effect of vitamin C and E supplementation at training. A reduction in maximal blood lactate concentration during an incremental test in endurance athletes was found after the supplementation [64]. Furthermore, an increase in aerobic power was found after three weeks supplementation of vitamins C and E [65].

## 7. The Effects of Vitamin C and Vitamin C and E Supplementation during Exercise in Two Distinct Groups: The Young and the Elderly

There are only a few data concerning the dependence of antioxidant supplementation on the age of the subjects during exercise training. There are some speculations that antioxidant supplementation can exert different effects in young and aged individuals. The concentration of vitamin C in male adolescents was positively associated with cardiorespiratory fitness level [66]. In the study of Suboticanec-Buzina et al. vitamin C supplementation resulted in a significant increase in VO_2_max in adolescents with initially lower values [67]. In contrast, other studies performed in young adults showed that vitamin C was not associated with cardiorespiratory fitness indices [68]. In young women, chronic vitamin C and E supplementation impaired strength-training-related improvements of body composition, such as a decrease in fat mass and an increase in fat free mass [39].

In the elderly population, Japanese authors found that the plasma vitamin C concentration positively correlated with handgrip strength, length of time standing on one leg with eyes open, and walking speed, and inversely correlated with body mass index [69]. Yet, Stunes et al. showed that high doses of the antioxidant supplementation may constrain the favourable skeletal benefits of 12 weeks of resistance exercise in healthy elderly men [40]. In another study, the supplementation had no effect on the increase of lean mass in trunk and arms, and muscle thickness of elbow flexors during strength training in the elderly [41]. However, Mason et al. revealed improvement in vascular function during exercise when vitamin C was implemented, which was especially apparent in a group of older adults [70]. Ryan et al. observed that vitamin C and E supplementation improved indices of oxidative stress associated with repetitive loading exercise and aging. In aged (but not young) rodents, administration of vitamins C and E improved the positive work output of muscles [49].

The effects of vitamin C supplementation in the elderly during exercise are not clear, though there is some more evidence of more improvement in this group of subjects than in the young ones. This difference may be partially explained by low-level inflammation, which often takes place in aged muscles, possibly due to a depletion of antioxidants, with accompanying increased oxidative stress in these muscles [71], and age-related muscle fiber loss leading to the development of sarcopenia. As a result, it seems that an aged muscle may benefit more from antioxidant supplementation. Furthermore, older individuals are prone to vitamin C deficiency. In this situation supplying the deficient agent may lead to positive effects, as in this case vitamin C levels can increase many fold [56]. However, the distinct effects of vitamin C administration on physical activity in the elderly has yet to be studied further.

In the reviewed literature it has very often been observed that vitamin C or vitamins C and E administration before or after exercise training blunts the training-induced adaptations to higher loads of oxidative stress. It is a paradoxical effect of the supplementation, which calls for caution when using these antioxidants during training. So far, the gathered evidence indicates that especially high doses of vitamins C and E are not advised (though often used by young athletes or people who undertake physical activity on regular bases). Perhaps a little more effort should be put into providing information on the antioxidant supplementation during exercise that is already obtained from research to the actively training population.

## 8. Dietary Nutrients, Sleep and Sleep Disorders

Health supplements represent a vibrant market, not only in the United States, but also around the world. The vitamin segment is expected to hold the largest dietary supplement market share. This phenomenon is possible thanks to the increase in health awareness of consumers of all age groups and their belief that these products produce health enhancement.

Studies have shown that people use dietary supplements for wellness and wellness-related reasons: disease prevention, improving immune function, energy, memory and concentration, but also enhancing sleep health [72].

Sleep is a basic requirement for infant, child, and adolescent health and development. In addition to nutrition and exercising, sleep is a critical determinant of human physical and mental well-being [73].

To have a restorative effect on the body, sleep must be of adequate quality and duration, which means that the following aspects should be taken into account: the total amount of sleep obtained per 24 h, the ease of falling asleep and returning to sleep, the placement of sleep within the 24 h day, the ability to maintain attentive wakefulness, and, finally, the subjective assessment of “good” or “poor” sleep [74].

The association of many different nutrients with sleep symptoms, such as difficulty falling asleep [75], sleep maintenance difficulties [75], poor sleep quality [76], non-restorative sleep and increased daytime sleepiness [75,77], duration of sleep [78] have been studied mostly in small trials or cross-sectional studies conducted in healthy adults.

Since vitamins and minerals are the most frequently consumed supplements among people today, their connection with sleep has also been the subject of intense research, and there is scientific evidence of an association between some vitamins and sleeping disorders.

According to the *International Classification of Sleep Disorders* [79], there are six major categories of sleep disturbances: insomnia, sleep-related breathing disorders, central disorders of hypersomnolence, circadian rhythm sleep–wake disorders including jet lag and shift work, sleep- related movement disorders, and parasomnias. Among them, obstructive sleep apnea, insomnia, and restless legs syndrome attract the greatest attention as a major public health issues. Sleep disorders not only impair the quality of life, but also increase the risk of other diseases, with deleterious health consequences, like hypertension [80], stroke, type 2 diabetes, obesity [81], cardiovascular disease and arrhythmias [82,83], impaired immune functioning [84], mood disorders [85], dementia [86], and neurodegeneration [87].

Scientists have also proven that insufficient sleep and untreated sleep disorders are associated with an elevated risk of culpable involvement in motor vehicle accidents [88,89].

As a major public health issue, the underlying causes of inadequate sleep quality and quantity are important to identify as well as developing the strategies to improve sleep-related health.

In this part of the present review we cite studies analyzing the link between antioxidant vitamin C and sleep outcomes, as shown in Table 3.

## 9. The Association of Vitamin C with Sleep Duration

About one third of our lives is spent asleep [105]. Sleep has to be of adequate length to have a restorative effect on the immune system and endocrine system, and to facilitate the recovery of the nervous system. The National Sleep Foundation guidelines advise that healthy adults (aged 26–64) need between 7–9 h of sleep per night; 7–8 h of sleep is recommended for elderly (over 65) [106]. Insufficient sleep, defined as fewer than 6 h for adults and fewer than 5–6 h for older adults, is associated with physical, physiological, and psychological impairments [106]. Prolonged sleep duration, defined as more than 9 h of sleep, is significantly associated with mortality, incident diabetes mellitus, cardiovascular disease, stroke, coronary heart disease, and obesity [107] Grandner et al., based on the data obtained from the 2007–2008 National Health and Nutrition Examination Survey (NHANES), demonstrated that both short and long sleep durations are linked with intake of several dietary nutrients. Analyses included adults aged 18+, and sleep duration, characterized as very short (<5 h per night), short (5–6 h per night), normal (7–8 h per night), and long (≥9 h per night), was estimated from survey items. Information about physical activity, overall diet, vitamins, and minerals intake was also assessed. The results of this analysis clearly showed that individuals with short sleep duration (5–6 h) demonstrated the lowest intake of lutein, zeaxanthin, selenium, and vitamin C, as the largest contributor of unique variance. There is a growing body of evidence linking free radical formation and oxidative stress with sleep and sleep-related disorders, and the significant association observed by Grandner et al. may suggest that an antioxidant, vitamin C, is important for sleep health [78]. 

These findings were further supported by Ikonte’s group, who investigated the relationship between short sleep and not only micronutrient intake, but also their inadequacy in U.S. adults aged 19+, based on the NHANES 2005–2016 database [90]. They also confirmed that adults reporting short sleep had inadequate intakes of vitamin C. Interestingly, Ikonte et al. found gender differences in the number of nutrients with insufficient intake, because an association between short sleep and inadequacy in several micronutrients, including vitamin C, was observed for adult women only, across all age groups (19–99) [90].

Another group, using the NHANES 2005–2006 database, analyzed the contribution of inflammation, oxidative stress, and antioxidants to the relationship between the duration of sleep and cardiometabolic health markers, since both short and prolonged sleep are risk factors for metabolic syndrome and cardiometabolic dysfunction [91]. The interrelationship between sleep duration and several serum measured factors related to inflammation (CRP), oxidative stress (GGT), and antioxidant status (bilirubin, carotenoids, uric acid, vitamins A, D, E, and C) was determined. What they discovered is that adequate sleepers (7–8 h per night) had optimal level of vitamin C and other antioxidants, as well as inflammation and oxidative stress profiles. Short (5–6 h per night) and very short (fewer than 4 h per night) sleep duration was associated with significantly lower vitamin C levels, together with vitamin D and carotenoids.

This concurs well with the observation made by Beydoun et al. that short or even very short sleep duration is linked to lower serum levels of vitamin C and other studied antioxidants, compared to normal sleep length [108].

Moreover, Kanagasabai et al. identified vitamin C as a modest mediator of the sleep duration-metabolic syndrome MetS relationship [91]. The results of this work support the suggestion that people with higher levels of vitamin C have better sleep health than those with lower levels. It is very likely, therefore, that adequate sleep duration is associated with optimal inflammation, oxidative stress, and antioxidant profiles, while sleep disorders are connected to elevated inflammation and oxidative stress levels and reduced antioxidant levels.

It is worth noting that vitamin C and other antioxidants, as well as inflammation factors, turned out to be significant mediators of several sleep duration–cardiometabolic health relationships only in women, which confirms the observations made earlier by Johnston et al. and Beydoun’s group [109,110]. It can be deducted that maintaining proper eating habits and sleep hygiene can especially improve women’s cardiometabolic health.

The results of the previous research were further supported by Noorwali et al., who performed the first nationally representative study based on data available for 1692 UK adults, to examine the connection between sleep duration and fruit/vegetable intakes and associated plasma biomarkers [92]. The participants, sleeping 7–8 h per day (reference sleep period) and having the highest intake of fruits and vegetables, were found to have higher levels of plasma vitamin C than short sleepers (fewer than 7 h per day). The major dietary sources of vitamin C are fruits and vegetables, and their intake is positively associated with plasma antioxidants. Interestingly, people characterized by long sleep periods (more than 8 h per day) had higher plasma vitamin C levels than reference sleepers. That could be explained by the use of varied diets or the discrepancy between the used biomarkers, measuring long-term dietary intake and the diet intake, which were assessed by a four-day food diary.

All of the above observations demonstrate that there is a strong relationship between vitamin C and the duration of sleep—both increased inadequacy and decreased serum levels of this essential antioxidant are associated with short sleep.

However, it should be noted that many factors can contribute to insufficient sleep duration; one of them is bedtime behavior, which was unknown in the case of surveyed participants.

It should also be emphasized that studies reviewed in this section provided only retrospective data correlating low vitamin C levels with short sleep duration, based on self-reported sleep duration and dietary intake. Despite demonstrating strong associations, as well as large, nationally representative groups studied and validated methods used, the weakness of these studies is that they are “retrospective in design”, and, as such, have some limitations to the interpretation of the results.

Is it important to optimize micronutrient status for sleep health benefits? Whether raising antioxidant status improves sleep duration remains an open question.

## 10. Vitamin C and Sleep Quality

Not only is it the duration of sleep that may be affected by vitamin and mineral intake or lack of these substances. There are studies that have shown the connection between vitamins and poor sleep quality. Poor quality sleep that does not relieve fatigue is usually associated with greater daytime sleepiness, which has important implications for health. Yet most of the literature data about the relationship between sleep and micronutrients focuses on sleep duration, not sleep disturbances.

A potential role of vitamin C as a contributor to restorative sleep was reported by Grandner and co-workers. The association between several sleep symptoms, like difficulty falling and staying asleep, non-restorative sleep, and daytime sleepiness, with the intake of specific nutrients was analyzed. The presented results are based on data of the National Survey conducted in 2007–2008 (NHANES) with the participation of 4548 adults aged 18 years. The most important findings from this study suggest that reduced intake of vitamin C can be associated with non-restorative sleep [75].

Insomnia is the most prevalent sleep disorder, occurring in a transient form for most people. It becomes chronic when the inability to attain adequate sleep lasts for at least three nights per week, for three months or longer [111]. Defined as a difficulty getting to sleep or staying asleep, and characterized by nonrestorative or non-refreshing sleep and waking up too early, it has serious consequences, including daytime impairment or distress, sleepiness, lethargy, and a general feeling of being unwell. These consequences have an economic costs.

Two main categories of insomnia include primary insomnia, unrelated to any underlying medical or psychiatric disorders, and insomnia associated with somatic or psychiatric disorders, most often anxiety, depression, stress, or specific sleep disorders [2]. Depression is a symptom of many vitamin deficiencies, and one of them is vitamin C, known for its antidepressant effects. Vitamin C is a well-established regulator of neurotransmitter biosynthesis. It serves as a cofactor for dopamine β-hydroxylase in the conversion of dopamine to norepinephrine (NE), which plays an important role in the regulation of mood. Chronic lack of vitamin C leads to decreasing NE levels [112]. Vitamin C is also a cofactor for the tryptophan-5-hydroxylase required for the conversion of tryptophan to 5-hydroxytryptophan in serotonin production, the deficiency of which contributes to depression [113]. It needs to be highlighted that depression [114], anxiety disorders [115], and psychosocial stress [116], which are psychological causes underlying insomnia, are associated with oxidative damage. Both preclinical and clinical evidence have demonstrated the beneficial effects of ascorbic acid supplementation on stress-related diseases such as depression and anxiety, although the neurobiological mechanism responsible for its neuroprotective properties is not fully understood [117].

Inadequate sleep is a substantial public health problem regularly affecting people worldwide. Although the first report linking insomnia with vitamin C was published in 1943, the direct link between this antioxidant vitamin and insomnia is still unclear and the results contradictory [118]. It is therefore important to understand whether insomnia can be caused by a deficiency or an excess of vitamin C. It is very likely that vitamin C supplementation, which produces an antidepressant effect and improves mood, relieves symptoms of insomnia.

Lichstein et al., looking for sleep-promoting and sleep-inhibiting effects of common vitamins (A, B, C, E, niacin), found that the use of a combination of vitamins, in the form of multivitamins or multiple single vitamins, resulted in disturbed sleep in some individuals and a higher rate of insomnia when compared to non-users [119].

These findings are not consistent with the discovery made by Matsuura and colleagues, who studied the relationship between insomnia symptoms and nutritional adequacy among Japanese adults. In a cross-sectional study based on data from a nationwide population survey conducted in 2013 they found that men with moderate to severe insomnia symptoms had an inadequate intake of vitamin C, together with vitamin B and foliate. In contrast to men, no such relation was found in women, suggesting a sex-specific relationship [93]. What is worth emphasizing is that the studied population had a low percentage of overweight individuals or current smokers, so the results obtained should be interpreted as results in populations with healthy lifestyle behaviors.

A growing number of clinical researches are showing that high-dose vitamin C might benefit the sleep health of cancer patients. Several studies have indicated that intravenous (IV) vitamin C alleviates some cancer and cancer-therapy-related symptoms and side effects, among other sleep disorders. Yeom et al. carried out a prospective study that included 39 cancer patients, who were given an IV administration of 10 g vitamin C (twice with a three-day interval) and an oral intake of 4 g vitamin C daily (for a week). The patients reported improvement in the quality of life, all functions, and some symptoms, with, among others, significantly lower scores for sleep disturbances [94].

Vollbracht and co-workers investigated the efficacy of IV vitamin C administration on the quality of life of breast cancer patients in a retrospective cohort studies. Patients were treated with 7.5 g vitamin C once a week for a minimum of four weeks, together with standard treatment. When compared with controls, significantly decreased sleep disorders, among other symptoms, were observed during chemo-/radiotherapy and the after-care phase [95].

Valuable observations have been made by Takahashi and co-workers examining the effects of high-dose (25–100 g/session, twice a week) intravenous vitamin C therapy, in addition to oral vitamin C doses of 2–4 g daily, on the quality of life of patients with advanced cancer. Insomnia, which is one of the cancer- and chemotherapy-related symptoms, was significantly improved within as little as one month after initiating vitamin C therapy [96].

Also a case study of a 45-year-old female diagnosed with breast cancer proved the effect of vitamin C therapy. Significant improvement in insomnia following IV vitamin C (50 g/session, twice weekly) administration for four weeks was observed [97].

Another case study, carried out by the same group, confirmed the effect of IV vitamin C treatment. A terminal cancer patient with angiosarcoma, after administration of 30 g vitamin C daily for one week, reported complete cessation of insomnia [98].

Oxidative stress is well known as a major etiological factor in the development and progression of cancer. Moreover, both radiotherapy and chemotherapy contribute to the increased production of free radicals. Oxidative stress is also increased in insomnia, due to oxidant–antioxidant imbalance. Vitamin C, being able to scavenge free radicals and reactive oxygen species, may decrease some of the symptoms related to cancer and cancer therapy, like weakness, fatigue, insomnia, or other sleep disturbances.

Vitamin C is needed to support many biological functions, and numerous studies confirm its low plasma levels in cancer patients, probably due to increased utilization of ascorbate during inflammation and oxidative stress. As Mayland et al. showed in their research, 30% of 50 patients with advanced cancer had plasma vitamin C deficiency and much shorter survival, highly correlated with plasma vitamin C concentration [120].

In none of the studies was patient vitamin C status at study entrance reported. The measurement of patient vitamin C status, not only at baseline, but also following the administration, should be taken into consideration—little is known about the actual concentration of this vitamin after injection. Based on pharmacokinetics data in healthy humans, plasma concentrations from oral dosing never exceed 250 µM, while intravenous doses produce 20 mM [121].

Supplementation may improve some aspects of a cancer patient’s quality of life and decrease multiple aspects of cancer-related fatigue because it relieves symptoms caused by the state of chronic vitamin C deficiency in these patients. On the other hand, vitamin C might act as an anti-cancer drug, acting selectively on cancer cells without affecting normal cells [122,123]. For this purpose, high doses of vitamin C, unachievable through oral administration, need to be administered intravenously. In some of the performed studies the authors declared that the administration of high doses of IV vitamin C restored vitamin C plasma levels [95,96]. In other cases improvement of quality of life (QOL) was achieved without reaching blood vitamin C concentrations high enough to exert anti-tumor effects [94], which indicates the existence of other mechanisms of action of IV vitamin C.

Consistent evidence exists that intravenous vitamin C can improve a cancer patient’s quality of life and decrease multiple aspects of cancer-related fatigue, but there are also important issues, like optimum dose used, inclusion of the placebo control group, and the measurement of some biomarkers of oxidative stress and inflammation [124].

## 11. Vitamin C and Obstructive Sleep Apnea

Obstructive sleep apnea (OSA) is a sleep disturbance and breathing disorder characterized by excessive daytime sleepiness, repetitive partial or total upper airway obstruction during sleep, snoring and hypoxemia. OSA predisposes one to cardiovascular disease and cerebrovascular events, independent of traditional risk factors, and increases morbidity and mortality in patients [125]. Redox imbalance, in addition to inflammation, has been proposed as an underlying mechanism for the symptoms characteristic of OSA syndrome (OSAS). During an episode of apnea, the oxygen saturation falls and the re-oxygenation occurs as the breathing resumes. The repeated hypoxia–re-oxygenation cycles result in oxidative stress, which is further enhanced by increase in free radical production caused by sleep deprivation per se [126]. Numerous studies have been conducted to evaluate the relationship between oxidative stress and Obstructive sleep apnea syndrome (OSAS). The vast majority of them suggest the connection between the increase in the stress and the disease, showing a strong correlation between the rise in oxidative stress markers and the fall of O_2_ saturation, the loss of antioxidant capacity of the patient, and the dependence of these factors on the severity of the disease [127,128,129,130,131,132,133,134].

What is interesting is that data published by the group of Ahiawodzi suggest that oxidative stress may be more associated with sleep-disordered breathing among women, not men [135].

Not only does an increase in ROS levels, but also a decreased antioxidant status contribute to the symptoms of OSA, as reported by Barcelo et al. [136]. However, some authors have denied the role played by oxidative stress and decreased antioxidant status in sleep apnea [137,138]. While most of the scientific evidence confirmed that sleep apnea is an oxidative stress disorder, it is worth asking if antioxidants improve sleep quality in OSAS patients.

Singh et al. tested this possibility in OSAS patients who were administered continuous positive airway pressure (CPAP) therapy for two nights, followed by oral intake of vitamin C (100 IU BD) and vitamin E (400 IU BD) for 45 days [100]. According to the authors, antioxidant treatment improved the quality of life for all patients with OSAS. Better still, comfortable sleep, a decrease in Epworth sleepiness, reduction in the number of apnoeic episodes and longer duration of sleep stages 3 and 4 were observed following vitamin C and E therapy. Also, the optimal pressure of CPAP was lowered and apnea-hypopnea index (AHI) was significantly decreased. The above results by Singh and co-workers supported the hypothesis that oxidative stress is an underlying mechanism for sleep disorder in OSAS patients. Intake of antioxidant vitamins C and E probably removes the inhibition caused by oxidative stress to the excitatory motor neuronal discharge to upper airway dilator muscles. A decrease in daytime sleepiness was explained by the authors as a consequence of the elimination of the damaging effects of oxidative stress on wake-promoting cells of the brain by antioxidant vitamins.

A rat experimental model of obstructive sleep apnea was used by Celec’s group to analyze the effect of antioxidant vitamins on markers of oxidative stress. Animals treated with the combination of vitamins C and E showed a lower concentration of advanced products of protein oxidation, in comparison to non-treated rats. These results clearly proved that these antioxidant vitamins are effective in alleviating oxidative stress, which contributes to the pathogenesis of cardiovascular complications [139].

Although experiments on animal models have limited benefits for human pathology, they indicate that vitamin C, with minimal risk of complications, can be considered as an alternative therapy for OSA patients, especially those with low tolerance to CPAP treatment [140].

This observation takes on particular significance in the context of recent findings reported by Campos-Rodriguez and co-workers that CPAP therapy does not improve biomarkers of antioxidant activity and inflammation in women with obstructive sleep apnea, compared to a non-treated control group. The study was conducted on 247 women with moderate to severe OSA treated with CPAP for 12 weeks. There is a rationale for expecting that CPAP, which improves many aspects of quality of life and daytime sleepiness, would also have a beneficial effect on diverse circulating biomarkers [141].

Researchers have been trying to determine the most potent antioxidant agent to be used to treat obstructive sleep apnea by minimizing the effect of oxidative stress. As an antioxidant therapy for OSA patients, vitamin C, next to N-acetylcysteine (NAC), was shown to be beneficial and effective in reducing harmful oxidative consequences of obstructive sleep apnea [140].

Vitamin C is also known to play a supportive role in preventing endothelial dysfunction, which is the first sign of various diseases associated with oxidative stress, including hypercholesterolemia, diabetes mellitus, hypertension, or congestive heart failure. On the other hand, scientific evidence exists that many diseases are characterized by cellular ascorbate deficiency as a cause for endothelial dysfunction (e.g., early atherosclerosis, sepsis, smoking, and diabetes) [142].

Grebe et al. tested the hypothesis that endothelial dysfunction associated with obstructive sleep apnea is also linked to oxidative stress [99]. For this purpose intravenously administrated 0.5 g of vitamin C was used and brachial artery flow was measured by ultrasound, before and after antioxidant injection in OSA patients. Such treatment acutely reversed endothelial dysfunction, presumably by decreasing the amount of circulating free oxygen radicals, in addition to restoring NO levels. Ascorbate has been shown to improve vascular function in patients with signs of endothelial disfunction. It does so by increasing bioavailability of NO through its interaction with tetrahydrobiopterin (BH_4_) (and maintaining BH_4_ in its reduced state) and endothelial nitric oxide synthase (eNOS) [143].

The findings of this study were further confirmed by Buchner’s group, who demonstrated in OSA patients that endothelial dysfunction of forearm microvasculature is also improved by the intra-arterial application of antioxidant vitamin C (25 µg/min) [101]. These studies confirmed that OSA impairs microvascular endothelial function to a degree that correlates with the severity of the disease. OSA-related hypoxia not only promotes oxidative stress, but also impairs blood vessel functioning, which can be reversed by the use of antioxidant strategy.

Both oxidative stress and impaired endothelial function are key cardiovascular risk factors, also involved in the pathogenesis of atherosclerosis. Vitamin C is not only a potent antioxidant, but also improves the endothelial health of sleep apnea patients to levels seen in people without sleep disorders. The use of free radical scavenger vitamin C should be explored for the treatment of OSA-related diseases.

## 12. The Effect of Vitamin C on Restless Legs Syndrome

Restless legs syndrome (RLS) is a neurological sleep disorder associated with disturbed sleep. RLS is classified as a movement disorder, characterized by a nearly irresistible urge to move, mostly the legs, which often accompanied by uncomfortable or unpleasant feelings and a sleep disorder, since symptoms may become more severe during the night. The most common problem caused by RLS affecting a patient’s sleep is difficulty initiating sleep. Disruption of sleep quality and length not only has an impact on a patient’s daily life and job, but also contributes to depression and anxiety [144].

RLS is a common disorder in hemodialysis patients. Chronic inflammatory status and the oxidative stress were proposed as significant contributing factors in RLS pathophysiology in patients undergoing hemodialysis [145]. That is why the efficacy of antioxidants in reducing the severity of RLS in patients who were under regular hemodialysis was assessed by different research groups.

Sagheb et al. tested for this purpose vitamin C tablets (200 mg) and vitamin E capsules (400 mg), and their combination [102]. After eight weeks of antioxidant therapy the intensity of the disease was measured using the International Restless Legs Scale (IRLS). These studies showed the effectiveness of vitamin C, vitamin E, and their combined therapy, which was proven by decreased IRLS sum scores. The authors concluded that the antioxidant properties of vitamins C and E contributed to the reduction of RLS symptoms.

Similar positive effects were obtained by Rafie et al. who evaluated the effect of eight weeks supplementation with vitamin C tablets (250 mg daily) in hemodialytic patients [103].

The impact of vitamin C on restless legs syndrome was further studied by Dadashpour et al. [104]. Hemodialysis patients were treated intravenously with vitamin C (500 mg/5 cc) following dialysis session, three times a week for eight weeks. The Pittsburg Sleep Quality Index and the Likert scale were used to assess the existence and intensity of sleep disorders and restless legs syndrome, respectively. The authors reported that patients receiving intravenous vitamin C declared significant reduction in RLS (*p* = 0.001) and remarkable statistical improvement in subjective sleep quality, the time of falling asleep, sleep latency, and daily dysfunction (*p* = 0.001), when compared with those treated with normal saline. Based on the results obtained, antioxidant vitamin C was suggested by the authors to be a simple and inexpensive potent solution for hemodialytic patients, improving sleep quality and decreasing restless legs syndrome. Moreover, vitamin C is well tolerated by these patients and its use has no adverse side effects that accompany the use of other medications, like popular dopamine agonists.

## 13. Conclusions

In this study, we summarize the existing evidence for the complex relationships between vitamin C and two physiological states, sleep and physical activity.

The importance of appropriate sleep and physical activity is extraordinary and supported by a significant body of research. Additionally, these two phenomena, though opposing in nature, have a positive influence on each other. Ascorbic acid is well known for its enormous health benefits and needs to be consumed on regular basis to prevent deficiency. The available evidence indicates that vitamin C is important for sleep, not only in healthy people, but also in patients with cancer. Vitamin C is also necessary for many metabolic reactions and antioxidant protection during exercise. Our work has led us to the conclusion that supplementation with vitamin C has a contradictory effect on sleep quality and physical activity. By increasing the consumption of this antioxidant, one can potentially help increase sleep duration, reduce sleep disturbances, relieve movement disorders, and decrease the dangerous effects of sleep apnea. On the other hand, caution is needed when using this vitamin as a supplement during physical training. No improvement, or sometimes even deterioration of physical performance, undesirable metabolic changes in the blood and muscles, and decline in antioxidant activity were often observed, especially when high levels of vitamin C were administered.

## Figures and Tables

**Table 1 nutrients-12-03908-t001:** Selected studies on the effect of vitamin C administration to subjects performing physical activity.

Type of Training	Participants Age (years)	Training Program	Duration	Intervention	Performance	Metabolic Changes	Antioxidant Funcztion	Studies
Endurance training	IG: 5 vs. CG: 9 untrained men Age: IG: 28 ± 1, CG: 31 ± 6	CAE; 3 × 40 min/wk	8 weeks	Vitamin C (1000 mg/d)	Trend for smaller improvement in VO_2_max			Gomez-Cabrera et al. [28]
	IG: 4 vs. 4 male rats Age: 86 ± 17 days	Wheel running	6 weeks	Vitamin C supplemented water (500 mg/kg for 3 weeks)	No change in DNA synthesis Decrease in protein synthesis rate	Decrease in mitochondrial biogenesis in the muscles	No difference in redox status and proteostasis	Bruns et al. [29]
	IG: 30 vs. CG: 30 men and women with metabolic syndrome Age: IG: 41 ± 6, CG: 42 ± 6	CAE: 30 min/d	12 weeks	Vitamin C (500 mg/d)	No difference in weight Decrease in waist circumference	Decrease in TG Increase in HDL-C	No difference in MDA	Farag et al. [30]
	IG: 8 vs. CG: 8 recreationally active men Age: IG: 21 ± 3, CG: 23 ± 2	HIIT: 4 × 30 min/wk	4 weeks	Vitamin C (1000 mg/d) vs. placebo	No difference in VO_2_max, running economy, and 10 km time trial	No difference in mean carbohydrate, fat oxidation rates		Roberts et al. [31]
Single exercise test	IG: 8 men vs. CG: 8 men Age: IG: 24 ± 1, CG: 24 ± 1	1 × 90-min intermittent shuttle-running test	_	Vitamin C (2 × 200 mg for 3 days after exercise)	No difference in muscle soreness, recovery of muscle function	No difference in CK activities and myoglobin		Thompson et al. [32]

All comparisons were made for groups supplemented with vitamin C and groups receiving placebo or no supplementation. IG—intervention group, CG—control group, CAE—continuous aerobic exercise, HIIT—high-intensity interval training, VO_2_max—maximal oxygen consumption, CK —creatine kinase, MDA—malondialdehyde, TG—triglycerides, HDL-C—high-density lipoprotein cholesterol.

**Table 2 nutrients-12-03908-t002:** Selected studies on the effect of vitamin C and vitamin E administration to subjects performing physical activity.

Type of Training	Participants Age (years)	Training Program	Duration	Intervention	Performance	Metabolic Changes	Antioxidant Function	Studies
**Endurance training**	IG: 20 vs. CG: 20 trained and untrained men Age: IG: 26 ± 3, CG: 26 ± 2	CAE, circuit training; 5 × 20–45 min/wk	4 weeks	Vitamin C (1000 mg/d) and vitamin E (400 IU/d)		Decreased PPARɤ, PGC-1a, PGC-1b, and insulin sensitivity	Decreased SOD, GPx	Ristow et al. [33]
	IG: 27 vs. CG: 27 trained and recreationally active men and women Age: IG: 25 ± 5, CG: 24 ± 6	CAE and HIIT: 5 × 30–60 min/wk	10 weeks	Vitamin C (1000 mg/d) and vitamin E (235 IU/d) vs. placebo	No difference in VO_2_max, 20 m shuttle run test	Decreased PGC-1a, COX-IV	Decreased uric acid No difference in SOD, GPx, GSH, HSP70	Paulsen, Cumming et al. [34]
	IG: 6 vs. CG: 5 recreationally active men Age: IG: 23 ± 1, CG: 22 ± 2	HIIT: 3 × 60 min/wk	4 weeks	Vitamin C (1000 mg/d) and vitamin E (800 IU/d) vs. placebo	No difference in VO_2_peak	No difference in CS, COX-IV Decreased TFAM	Decreased SOD	Morrison et al. [35]
	IG: 11 vs. CG: 10 recreationally active men Age: IG: 29 ± 5, CG: 31 ± 5	CAE and HIIT: 5 × 30–120 min/wk	12 weeks	Vitamin C (500 mg/d) and vitamin E (400 IU/d) vs. placebo	No difference in VO_2_max, LT	No difference in PPARɤ, PGC-1a, β-HAD, CS, glycogen concentration	No difference in SOD	Yfanti et al. [36]
**Resistance** **training**	IG: 14 vs. CG: 14 recreationally active men Age: IG: 26 ± 2, CG: 26 ± 1	RT: 2 ×/wk	4 weeks	Vitamin C (1000 mg/d) and vitamin E (400 IU/d) vs. placebo	No difference in muscle torque, muscle resistance to damage			Theodorou et al. [37]
	IG: 17 vs. CG: 15 recreationally active men and women Age: 20–45	RT: 4 ×/wk	10 weeks	Vitamin C (1000 mg/d) and vitamin E (235 IU/d) vs. placebo	No difference in lean mass. Decrease in strength increases (biceps curl) and protein synthesis			Paulsen, Cumming et al. [38]
	IG: 12 vs. CG: 11 untrained women Age: IG: 23 ± 2, CG: 23 ± 2	RT: 2×/wk	10 weeks	Vitamin C (1000 mg/d) and vitamin E (400 IU/d) vs. placebo	Decrease in total lean mass, deadlift strength, lunge strength		No difference in IL-6, MDA	Dutra et al. [39]
	IG: 17 vs. CG:18 men Age: 68 ± 6 year	RT: 3×/wk	12 weeks	Vitamin C (1000 mg/d) and vitamin E (235 IU/d) vs. placebo	Decrease in total lean mass, aBMD No difference in 1 RM	Decrease in IGF-1, leptin, adiponectin, resistin No difference in TNF-α		Stunes et al. [40]
	IG: 17 vs. CG: 17 untrained men Age: IG: 69 ± 7, CG: 67 ± 5	RT: 3×/wk	12 weeks	Vitamin C (1000 mg/d) and vitamin E (400 IU/d) vs. placebo	Decrease in lean mass, thickness of m. rectus femoris No difference in 1 RM leg extension, 1 RM leg press, 1 RM bicep curl			Bjørnsen et al. [41]
**Single exercise test**	14 physically active men Age: 21 ± 0.3	1 × 5 km continuous cycling test	_	Vitamin C (1000 mg/d), vitamin E (600 IU/d), α-lipolic acid (600 mg/d)	Decrease in power output, 5 km time, ventilation, economy, fatigue No difference in VO_2_	Increased blood lactate		Vidal et al. [42]

All comparisons were made for groups supplemented with vitamin C and vitamin E and groups receiving placebo or no supplementation. IG—intervention group, CG—control group, CAE—continuous aerobic exercise, HIIT—high-intensity interval training, RT—resistance training, VO_2_max—maximal oxygen consumption, VO_2_peak—peak, aBMD—areal bone mineral density, TFAM—mitochondrial transcription factor A, HAD—β-hydroxyacyl—CoA dehydrogenase, CS—citrate synthase, GPx—glutathione peroxidase, SOD—superoxide dismutase, MDA—malondialdehyde, COX-IV—Cytochrome c oxidase subunit IV, GSH—glutathione, HSP 70—heat shock protein 70, TNF-α—tumor necrosis factor α, IL-6—interleukin 6, IGF-1—insulin-like growth factor-1, PPARɤ—peroxisome proliferator-activated receptor ɤ, PGC-1a—peroxisome proliferator–activated receptor gamma coactivator-1 alpha, PGC-1b—peroxisome proliferator–activated receptor gamma coactivator-1 beta.

**Table 3 nutrients-12-03908-t003:** Example of studies linking vitamin C and sleep outcomes.

Sleep Outcome Variables	Study Design	Participants	Vitamin C Serum/Plasma Deficiency	Vitamin C Inadequate Intake	Vitamin C Intervention	Sleep Outcomes	Studies
**Sleep duration:** Short sleep	Cross–sectional study	n = 5587 adults aged 18+		+			Geandner et. al. [78]
	Cross–sectional study	n = 26,211 adults aged 19+		+ female only			Ikonte et al. [90]
	Cross–sectional study	n = 2079 adults aged 20+	+				Kanagasabai et al. [91]
	Cross–sectional study	n = 2612 adults aged 19–65					Noorwali et al. [92]
**Sleep disturbance:** Non-restorative sleep	Cross–sectional study	n = 4552 adults aged 18+					Grandner et al. [75]
**Sleep disorders** insomnia	Cross–sectional study	n = 1997 adults aged 18–69		+ men only			Matsuura et al. [93]
	Prospective study	n = 39 IG: terminal cancer patients			IV vitamin C (10 g 2×/week) Oral vitamin (4 g/d for one week)	insomnia	Yeom et al. [94]
	Retrospective study	n = 125 IG: 53 breast cancer patients CG: 72 patients			IV vitamin C (7.5 g 1×/week ≥4 weeks) vs. no vitamin C	Sleep disturbances	Vollbracht et al. [95]
	Prospective study	n = 60 IG: advanced cancer patients			IV vitamin C (25–100 g 2×/week for four weeks) Oral vitamin(2–4g/d)	insomnia	Takahashi et al. [96]
	Case study	n = 1 IG: breast cancer patients			IV vitamin C (50 g session 2×/week for four weeks)	insomnia	Carr et al. [97]
	Case study	n = 1 IG: terminal angiosaicoma patients			IV vitamin C (30 g/d for one week)	insomnia	Carr et al. [98]
	Single blind Randomnized Controled study	n = 20 IG: 10 OSA patients CG: 10 health subjects			IV vitamin C (0.5 g bolus injection)	Endothelial dysfunction	Grebe et al. [99]
Sleep apnea	Randomnized Controled study	n = 30 males IG: 20 OSA patients CG: 10 health subjects			CAPA for 2 nights Oral vitamin C (100 IU BD) Oral vitamin E (400 IU BD) for 45 days	Better sleep epowrth scal sleep stages 3 and 4 CAPA pressure	Singh et al. [100]
	Randomnized Controled study	n = 20 males IG: 11 OSA patients CG: 9 subjects w/o OSA			IA vitmain C (25 ug/min) CAPA for 6months	Endothelial dysfunction	Buchner et al. [101]
Restless legs syndrom	Randomized double-blinded placebo-controlled study	n = 60 IG: hemodialysis patients			Vitamin C tablets (200 mg) or Vitamin E capsules (400 mg) or Vitamin C tablets (200 mg) and Vitamin E capsules (400 mg) for 8 weeks	Severity of RLS	Sagheb et al. [102]
	Randomized double-blinded clinical trial	n = 45 IG: hemodialysis patients			Vitamin C tablets 250 mg for 8 weeks	Severity of RLS	Rafie et al. [103]
	Randomized double-blinded clinical trial	n = 90 IG: hemodialysis patients			IV Vitamin C (500 mg/ 5 cc 3×/week for 8 weeks )	Severity of RLS Sleep quality Time of falling sleep Sleep latency Sleep dysfunction	Dadashpour et al. [104]

IG—intervention group, CG—control group, IV—intravenous, IA—intra-atrial, OSA—obstructive sleep apnea, CPAP—continuous positive airway pressure, RLS—restless legs syndrome.

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
