# Peer review of "The Role of Vitamin C in Two Distinct Physiological States: Physical Activity and Sleep"

_nutrients, 2020, doi:10.3390/nu12123908_

Round 1

Reviewer 1 Report

Nutrients: The role of vitamin C in two distinct physiological states: physical activity and sleep

Thank you very much for the opportunity to review this interesting literature overview. The authors have summarized a substantial amount of literature on Vitamin C supplementation for physical activity recovery (?) in various populations, as well as vitamin C supplementation to improve sleep in general and in different physiological and psychological sleep disorders. However, I do have some concerns, which I highlighted below.

Is this a literature overview or systematic review? Please indicate in the title of the manuscript. The text itself suggests it is a literature overview, however, the tables indicate a systematic review approach. The latter is of greater scientific value, particular when looking at the length and number of included studies. Therefore, information on how the authors searched for the cited studies is of relevance.

Generally, I advice to rewrite the manuscript with a more focused approach. It remains unclear why the authors have chosen physical activity and sleep, as nowhere in the body of the text the two physiological states are linked (except for that they are both important). Why is it important to look at sleep and PA? What population or reader group will benefit from the provided review? The manuscript would make a larger contribution to the scientific community if authors would focus on a specific population for whom an improvement in physical activity or sleep would be beneficial:

  • Athletes
  • Sedentary
  • Diabetes
  • Obese
  • Elderly
  • Cancer patience
  • Metabolic syndrome
  • Cellular adaptations

After selecting a population (or more), specify the outcome. From the text it occurs as if you are mainly focusing on recovery improvement from (high intensity or long-endurance) exercise. This would significantly sharpen your focus and selection of studies. Please also state when Vitamin C was supplemented and how much (amount per dose/frequency). If you want to focus on Vitamin C only, then exclude studies in which vitamin C and E were studies simultaneously, as findings might be misleading when looking for Vitamin C effects only.

The authors state that they are focusing on sleep disorders, particularly OSA, insomnia and restless leg syndrome. When looking for beneficial effects of Vitamin C on these sleep disorders, first, the etiology and specific symptoms should be stated. Otherwise, the underlying effects remain unclear and questionable with regard to Vitamin C supplementation. Particularly insufficient sleep duration is often the result of bedtime behavior. Extended sleep of the cause of diseases. Insomnia is characterized by increased fatigue and not daytime sleepiness (authors falsely use both as synonyms). Again, authors should describe the general underlying causes of insomnia (psychological) and how scientific evidence suggests that vitamin C supplementation can be beneficial in this process. In turn, insufficient sleep impacts on nutritional behavior. OSA is often present in individuals with overweight. OSA itself results in hypertension and may lead to diabetes. This link was not made by the authors when reporting about the effects of Vitamin C supplementation in individuals with diabetes or metabolic syndrome in response to physical training. So again, manuscript lacks a precise focus of the target populations and why PA and sleep were chosen if there is no link between the two of them. The conclusion would have provided the opportunity to link the two of them and to give an outlook or take home message.

Minor: Check for up-to date literature: The introduction lacks literature sources and amongst others, the physical activity paragraph lacks up-to-date literature.

Author Response

Aleksandra Krol, Ph.D.                                              Lodz, 2020.12.10

Medical University of Lodz

Department of Experimental Physiology

Lodz 92-215

Mazowiecka 6/8

Dear Reviewer,

Thank you very much for your valuable comments and suggestions. Please find our corrected manuscript entitled: “The role of vitamin C in two distinct physiological states: physical
activity and sleep”. This manuscript was corrected according to the Reviewer remarks and we hope that it is now suitable for publication in Nutrients.

Answer to Reviewer:

Major points:

  1. L.120 (now L.157): The statements here were indeed contradictory. We must have accidently put the sentences in the wrong order and did not notice it. The sentence order is now corrected, which should result in more sense.
  2. L.275 (now L.292 and L.401): Thank you for your comment and the valuable information on intestinal absorption of vitamin C which we used in two sections of the article (5. Antioxidant and anti-inflammatory properties of vitamin C and E in exercise and 7. The effects of vitamin C and vitamin C and E supplementation at exercise in two distinct groups: the young and the elderly).
  3. L.352 (now L.480): Yes, we do agree that the studies reviewed in this section provide only retrospective data based on self- reported sleep duration and dietary intake, and as such can only provide correlations data. But the research cited prove strongly that very short and short sleepers consume less vitamin C. The mechanism that would explain the differences in vitamin C intake between sleep duration groups is unknown. The strong point of this research is large, nationally representative study (n=5587)  and that the method employed by NHANES to  assess dietary intake has been validated. Also, the association of vitamin C intake below EAR with short sleep was significant. The analyzed studies did not include college students, but showed the strong correlation between inadequate intake and short sleep across all age groups: 19-99. Many other factors can contribute to short sleep, but many of them were addressed as covariates. We also agree that ” The association of vitamin C with sleep duration” would be a more correct version of the heading.The heading has been changed and the weakness of the retrospective studies has been underlined.
  1. L.430 (now L.573): It needs to be highlighted that in none of the studies cited patient vitamin C status at study entrance was reported. Additionally, little is known about actual concentration of this vitamin after injection. The authors declared that the administration of high doses of IV vitamin C restored vitamin C plasma levels (Takahashi et al., Vollbracht et al.). In some of the performed studies improvement of QOL has been achieved without reaching blood vitamin C concentrations high enough to exert antitumor effects (Yeom et al.), which indicates the existence of other mechanisms of action of IV vitamin C.

We do agree, many molecular studies suggest that vitamin C can act at least in two opposite ways: anti- and pro-oxidant. Despite the vast knowledge in this field, human epidemiological studies and clinical trials do not determine the role of vitamin C supplementation in cancer prevention and therapy. The “pro-oxidant” activity of vitamin C is thought to predominate at higher doses, as demonstrated in murine models, but there is little evidence to indicate that the proposed “pro-oxidant” mechanism is occurring in oncology patients administered IV vitamin C.

  1. L.563 (now L.745): The relationship between vitamin C and NO was only mentioned, it is now underlined in the text.

Minor points:

  1. L.35 (now L.38): NREM and REM sleep have been defined.
  2. L.109 (now L.148): “Selen” has been changed to “selenium”.
  3. L.462 (now L.625): “Preclinical” has been changed to “clinical”.

With best regards,

Aleksandra Krol

Reviewer 2 Report

This review attempts to contrast different roles of vitamin C in physical exercise and sleep. Overall it is well written and informative, especially with the positive studies in sleep apnea and restless legs syndrome.

Major Points:

  1. L: 120: the negative effects of vitamin C noted don’t make much sense – a decrease in lipid peroxidation would seem a positive rather than a negative effect. Indeed, for glutathione peroxidase, this statement contradicts that on L. 125.
  2. L: 275: This is an important point. Supplementation of what are likely normal or even increased levels of circulating vitamin C may well have little effect on the levels. That is, uptake of vitamin C is closely regulated by intestinal transport on SVCT1, such that most studies show only a modest effect on levels, even several hours after ingestion. To be convincing, any study should show whether or not vitamin C levels were significantly affected by the supplement. The authors might consider looking through the cited articles for any that actually measured vitamin C. Since that is not likely they could cite the studies of human supplementation by Mark Levine at NIH on how much a certain dose of C will increase levels.

A related point made by the authors is that giving vitamin C to subjects deficient in vitamin C may have effects, because then the levels can increase many fold. This may have relevance to studies in older humans, who may be deficient.

My main request is for the authors to consider that increases in vitamin C levels are somewhat limited by uptake from the gut. Also, emphasize that there may be substantial differences in results if the subjects are low or deficient in the vitamin.

  1. L. 352: the studies reviewed in this section provide only retrospective data correlating low vitamin C levels with short sleep duration. The heading here, that low vitamin C has an effect on sleep, may not be correct. More likely, those with short sleep have low vitamin C because they are too busy or also have poor dietary intake – e.g. college students, 14% of whom are deficient in vitamin C. This heading should be changed and this caveat (weakness) of the respective studies should be emphasized.
  2. L. 441: the point in #3 above about intake is noted here – but still should be a caveat with the studies described L. 352.
  3. L. 430:  the IV supplement effects of vitamin C are substantial – again, note the huge increase in plasma levels with IV administration -even if not measured. Or if deficient, an increase to normal.

One other point:  the mechanism of the anti-cancer effect of IV vitamin C appears to relate to a prooxidant effect of the vitamin with an iron-contain protein in the tumor. This destroys cancer cells, so that a decrease in tumor mass might contribute to improved sleep. Again, these are studies of Mark Levine. The authors don’t need to discuss this, since it is likely tangential to the vitamin C effects.

  1. L. 563: vitamin C is known from pre-clinical studies to preserve nitric oxide. It does this by reducing tetrahydrobiopterin, a key intermediary in NO biosynthesis.
  2. Conclusion: perhaps reiterate that 1) some studies may not have given enough vitamin C to volunteers or were negative because subjects weren’t deficient, 2) the IV studies in sleep clearly showed cause-and-effect.

Minor points:

  1. 35: Define NREM and REM sleep initially
  2. 109: selen should probably be selenium- or if not, define
  3. 462: preclinical usually means animal studies.

Author Response

Aleksandra Krol, Ph.D.                                                         Lodz, 2020.12.10

Medical University of Lodz

Department of Experimental Physiology

Lodz 92-215

Mazowiecka 6/8

Dear Reviewer,

Thank you very much for your valuable comments and suggestions. Please find our corrected manuscript entitled: “The role of vitamin C in two distinct physiological states: physical
activity and sleep”. This manuscript was corrected according to the Reviewer remarks and we hope that it is now suitable for publication in Nutrients.

Major points:

  1. Our paper is a literature overview. We have now added this information to the abstract.
  2. Thank you for your comment that we should provide more information on how sleep and physical activity are linked. We have now provided links of these opposing states in relations to the redox state of the body in Introduction, and then in the body of the text.
  3. Thank you for your comment that we should put more focus to our study and to specify the outcomes and populations. This comment was especially valuable. We have reorganized our review, rewritten the aims of the study, which helped to divide and specify the changes we wanted to observe in relation to physical activity. The aims result in separate sections of the article with separate outcomes. In each section of our work we specify what population we focus on.
  4. We have put the dosage and frequency of vitamin C and vitamin C and E of selected studies (most of the studies mentioned in this review) into the Tables
  5. Thank you for pointing out that the effect of studies on vitamin C and vitamin E may not be attributed to vitamin C only. We see that it was misleading. We have now divided the results of the studies into those on the effect of vitamin C only and vitamin C and E . Each section of the review focusing on physical exercise begins with the effect of vitamin C , and the second part of the section is on the effect of vitamin C and E. In order to make this division more visible we have now made two separate tables in relation to physical activity, i.e. Table 1 on the effect of vitamin C and Table 2 on the effect of vitamin C and E.
  6. We do agree that that many factors contribute to short sleep. Most of them were addressed as covariates in cited studies. Insufficient sleep duration is often the result of bedtime behavior, which can worsen sleep. But there is no information about specific eating habits, like food consumed and timing of the meal, sleeping time, shift-work or chronotype. Such limitation of the data results in an unknown bedtime behavior. The only information we have is that very short and short sleepers were less educated, with lower income, Black/African Americans, with higher BMI and more physical activity (Grandner et al.). Studies reviewed in this section provide only retrospective data, but  they use large nationally representative sample, which is one of the strongest side.
  7. The brief description of insomnia was included in the text. More detailed characteristic, with symptoms and general underlying causes has been added. Also, the link of vitamin C with the psychological causes of insomnia, as well as benefits of vitamin C supplementation have been explained.
  8. Sleepiness is one of the consequences of chronic insomnia and only in this context is the term used (L), but not as a synonym.

Minor:

In Introduction sources were added.

Some of the outdated literature was removed from the „Physical activity” paragraph.

With best regards,

Aleksandra Krol